# Career Development of Adapted Sports Coaches: Systematic Review of Qualitative Evidence Literature

**DOI:** 10.3390/ijerph18126608

**Published:** 2021-06-19

**Authors:** Pedro Pires, André Ramalho, Antonio Antúnez, Helena Mesquita, Sergio J. Ibáñez

**Affiliations:** 1Research Group in Optimization of Training and Sports Performance (GOERD), University of Extremadura, 10005 Cáceres, Spain; pedroruiinespires@gmail.com (P.P.); sibanez@unex.es (S.J.I.); 2Sport, Health & Exercise Research Unit (SHERU), Instituto Politécnico de Castelo Branco, 6000-266 Castelo Branco, Portugal; andre.ramalho@ipcb.pt (A.R.); hmesquita@ipcb.pt (H.M.); 3Centro Interdisciplinar de Ciências Sociais da Universidade Nova (CISC.NOVA), 1070-312 Lisboa, Portugal

**Keywords:** coach, adapted sport, career development, training path

## Abstract

This study aims to synthesize and understand the qualitative and empirical evidence previously published on adapted sports career development. This systematic literature review also aimed at developing a conceptual model of the training and career path of adapted sports coaches. The research was carried out through the following databases: PubMed, Erid, EBSCO, Web of Science and Scopus. We selected nine qualitative studies according to the specific eligibility criteria. Data extraction was carried out independently by different authors, including the assessment of the methodological quality of the articles. We developed a conceptual model composed of four descriptive subjects (adapted sport option; perception of training athletes in adapted sports; development of the adapted sports coach’s learning; adapted sports coach reality) about the adapted sports coach’s pathway and added three analytical subjects about the difficulties, opportunity and reality of adapted sports. The conceptual model suggests inclusion of practical activity programs, the specificities of related contents with the adapted modalities, and observation in a real context are fundamental.

## 1. Introduction

Sports are an activity that follows a structure that is perfectly defined by technical, tactical and psychosocial rules and dimensions, where coaches play a fundamental role for athletes who view them as leaders and experts [1]. In this respect, the coach’s role is to optimize and maximize the potential of the team and/or the athlete [2]. In the context of adapted sport, coaches have to adapt their knowledge and skills using a combination of innovation, intuition and creativity in order to adjust training to the specific needs of the athletes [3].

Studies carried out on coach’s training have been an important ally in the dissemination of knowledge, namely what coaches need to obtain in order to effectively perform the job [4], but the limitation of investigations of primary studies and revision in the adapted sport (AS) area is evident [5].

Irwin et al. [6] defend the idea that the identification of the trainer’s knowledge is the conceptual mechanism for their development and is a fundamental process in their evolution. For the trainers, the most significant sources of knowledge are practical learning (mentoring), since it helps them to reflect on the practical aspects of the job. Werthner and Trudel [7] further specify the subject matter, reinforcing the concept that studies on trainers’ learning processes are fundamental elements in developing more effective work, concluding that the process is idiosyncratic, having to consider both formal and informal knowledge.

Thus Ibáñez et al. [8] stress the need for further research to delve more deeply into the theory and practice of the coach’s role. In this context, identification of the knowledge of coaches [6] and their learning process [7] are paradigms which, according to the authors mentioned, lack greater scientific depth and therefore present important concerns for the production of a systematic literature review.

In view of the abovementioned arguments, this research is revealing in both formal and informal contexts. From a formal perspective, Haegele et al. [9] conducted a study on research trends in adapted physical activity between the years 2004 and 2013, where they warned that the analyzed research lacks conceptual and theoretical structures. They identified in their review that research on adapted sport was characterized by descriptive non-intervention and group-design studies. They underlined the importance of these studies, as the scope of adapted physical activity should be considered by high-impact scientific sources to proliferate knowledge in the area of adapted sports. As for the informal relevance, the study is basic for both coaches and training institutions, since it is suggested that the contents of the training courses be reformulated to include additional resources [2].

Bentzen et al. [10] conducted a comprehensive review with the aim of enhancing the advancement of knowledge in this specific area by analyzing articles published between 1991 and 2018. The main objective of the study was to provide a broad perspective on the existing literature on adapted sports coaches. This specific research was concerned with characterizing the investigative design of the studies produced within the scope of adapted sports. These authors highlighted the bias of this research, which uses interviews to obtain information, and the prevalence of male coaches. In their study they identified that the three most frequently addressed topics in these studies are becoming a parasport coach, being a parasport coach, and general parasport coaching knowledge. They found that the formal training of coaches is limited, expensive and lacks specific content for coaching in adapted sport. In addition, they considered it important to develop a conceptual model of the effectiveness of coaching in adapted sport.

Given this prerogative and as a way to obtain a holistic perspective on the coach’s knowledge and relationship with the athletes, the model by Bloom et al. [11] suggests the 3 + 1 Cs [12,13], since this model allows the coach to adjust to the individual needs of the athletes. In conceptual terms, this model aims to measure the relationships between coaches and athletes on an emotional, cognitive and behavioral level. The four constructs underpinning this model are closeness, commitment, complementarity, and co-orientation [13]. Co-orientation is the factor that links the three previous constructs, as it is defined as the shared knowledge and understanding between coaches and athletes.

As far as is known, there is little research that deals specifically with the professional career of coaches working in adapted sport. Therefore, the aim was to make a systematic review to identify studies that analyze coaches in adapted sport. From this perspective, the lack of knowledge and specific training of professionals for dealing with disabled people [14,15] is an obstacle in the progression and development of the area of AS. This study aimed to synthesize and understand the qualitative and empirical evidence previously published about AS career development. In addition, this review also aimed at developing a conceptual model on the training and career path of adapted sports coaches (ASCs).

## 2. Method

This systematic literature review was developed from the guidelines of Enhancing Transparency in Reporting the Synthesis of Qualitative research (ENTREQ) [16]. In addition, the recommendations of Preferred Reporting Items for Systematic Reviews and Meta-Analyses (PRISMA) [17] were also followed.

### 2.1. Eligibility Criteria

The following literary research strategy was used: searching for studies in different scientific databases (Pubmed, Eric, EBSCO, Web of Science, Scopus) using keywords associated with the defined eligibility criteria (sports and disability and coach). The time period analyzed was up to February 2020. The eligibility criteria for the studies were developed from the Sample, Phenomenon of Interest, Design, Evaluation and Research type (SPIDER), especially recommended for the synthesis of qualitative evidence [18]. Thus, the systematic review obeyed the consensus regarding the research, since this procedure is used in observational studies and allows the reviewers to establish an agreement before the observation is carried out [19]. The criteria are represented in Table 1.

### 2.2. Literary Research

The following literary research strategy was used: searching for studies in different scientific databases (PubMed, Eric, EBSCO, Web of Science, Scopus) using keywords associated with the defined eligibility criteria (“coach”; “knowledge”; “adapted sport” “Parasport coach”, “Para sport”, “disability sport”). The keywords of the study were developed from three areas: (1) coaches or related terms; (2) knowledge of the coach, and more specifically (3) in the area of AS. The keywords used were searched for in combination with each other. In addition, the literature reference lists of selected studies were consulted for the verification of other relevant studies. Data collection took place from 16 October 2019 to 26 February 2020, including all existing documents in which the keyword combination was identified up to the closing date of the study.

### 2.3. Selection of Studies

The studies were inserted into the EndNote software (Thompson Reuters, San Francisco, CA, USA), and the duplicates were removed through the “duplicates” function. After this procedural phase, the studies were selected through the following stages [20]: in the first stage, two reviewers, independently, selected the studies based on their titles. In case of doubt about their relevance to the goal of the study, they were included for the next stage. In the second stage, the abstracts of the included studies were analyzed. Disagreements about the exclusion and inclusion of the studies for the subsequent phase were resolved through the mediation of a third reviewer. In the final stage, the studies were read in full by two reviewers. Discrepancies about the inclusion of studies for the synthesis of the final evidence were resolved by consensus [19] among the three reviewers who participated in the previous phases. The studies were included in the final synthesis considering the defined eligibility criteria.

### 2.4. Data Extraction

The data were extracted by two independent reviewers. The following categories were considered: study characteristics; design; objective; data analysis and results/conclusions. In this way, the data from the selected articles were inserted in a table prepared to summarize the obtained information [21]. In addition, the quotations from the participants that are transcribed in the selected studies were also extracted in order to establish a more general understanding of the data [22]. At this stage, the discrepancies about the extracted data were resolved through the mediation of a third reviewer.

### 2.5. Quality Assessment

The assessment of the methodological quality of the selected studies was carried out through The Critical Appraisal Skills Program (CASP) [23]. This instrument has 10 assessment items that are classified into: Yes; Can’t tell or No. Studies have been classified as “high quality” if they meet at least 8 of the 10 criteria, “medium quality” if they meet 5–7 of the criteria, and “low quality” if they meet 4 or less. The studies were independently evaluated by three reviewers. Disagreements concerning the classification of studies were resolved by the reviewers by consensus.

### 2.6. Data Analysis

A thematic synthesis [24] was carried out on the data extracted from the selected studies giving origin to descriptive topics. However, the synthesis was intended to go beyond a description of the data collected. From this perspective, analytical topics were developed from the descriptive ones that allowed the construction of new interpretations regarding the development of an ASC’s career, experimenting with the existing knowledge. At the formal level the thematic synthesis was developed in three phases. Peer debriefing was used to ensure the credibility of the data analysis [25]. Thus, during the investigation, periodic meetings were held with the research team to critically review the entire data analysis process. The first phase corresponded to the coding of the subjects of the studies included, independently allowing the description, synthesis and establishment of key concepts of each study. After this stage, the descriptive subjects were developed. In the first instance, the primary analysis was very close to the conclusions drawn from the studies included, so it was necessary to develop analytical subjects that allowed the goal of the study to be achieved [24]. The subjects were developed with the aim of guaranteeing their exclusivity and exhaustiveness [26]. All the procedures described above were prepared in the first instance by the first author, who made the thematic map. In a second phase, the thematic map was independently reviewed by two different reviewers and the disagreements existing in the thematic synthesis were resolved among the three reviewers by consensus [19].

## 3. Results

### 3.1. Selection Process of Studies

Figure 1 presents an overview of the study selection process using the PRISMA flowchart. The initial search identified 49 titles in the database and 13 more relevant investigations were added for the review. A total of 62 records were identified. These data were exported to the reference database management software (EndNote), and any duplicates (47 references) were automatically deleted.

The remaining 47 articles were tracked according to the title and abstract, resulting in 34 studies being deleted from the database. In the final process the articles were fully read (*n* = 13) and four studies were excluded because they did not meet the inclusion criteria.

### 3.2. Characteristics of Studies

Table 2 characterizes the studies included in the revision, all of them being investigations using a qualitative matrix. Most (*n* = 7) are studies of Canadian origin, except for two, one relating to the USA and one to Australia. Regarding the characteristics of the studies, due to the particularities of the topic of the revision, information can be found on sample number (*n*); type of coach (TC); sport category (SC) and typology of disability (TD). The sample size varies between 1 and 45. It should be noted that four studies have exclusively men (m) trainers in the group of participants, four studies have participants of both genders and only one research has only women (f) trainers. All the studies took into account the selection of elite trainers and only two grouped elite trainers with beginners. In the sports aspects, the studies focused on individual (*n* = 3) and team (*n* = 5) modalities and one of the studies did not specify the modality. The type of disability is a subject which is not very characteristic, since a large part of the studies (*n* = 7) either did not specify the pathology or addressed more than one type of disability. The in-depth interview was the most commonly used method, based on structured and semi-structured interviews. The semi-structured interview was the most used method for data collection (*n* = 5), followed by both in-depth (*n* = 2) and structured (*n* = 2) interviews. In the analysis of the data, most investigations chose to use a thematic analysis (*n* = 8).

### 3.3. Assessment of the Study Quality 

Table 3 shows the results of the paper quality coding system proposed by CASP, which is a qualitative evaluation tool. The CASP has 10 questions, which focus on each methodological aspect of the study being analyzed [23]. Each question is evaluated with a “yes” if the analyzed aspect is evident; “can’t tell” if the analyzed aspect is not clear; “no” if it is not consigned. According to this evaluation, the articles were classified as high quality when they score between 10 and 8 points, medium quality 7 to 5 points and low quality 4 to 0 points. In the present research most studies are classified [23].

### 3.4. Thematic Analysis

Figure 2 formally demonstrates the process of designing the thematic synthesis which comprised four topics and 12 descriptive sub-topics. Furthermore, another three analytical topics and eight analytical sub-topics were established.

### 3.5. Descriptive Subjects

#### 3.5.1. Adapted Sport Option

This subject is related to the choice and not to the AS. The coaches who make the career choice have influential factors to make this decision, namely previous experience with people with disabilities, family and/or academic experience and interest in the area. In addition to the premises mentioned, personal experiences are revealed in the choice, as they include a number of important aspects such as personal skills, having been an AS athlete, volunteer experiences in the area and experiences in overcoming barriers throughout life. The lack of option for AS is related to contextual factors and change. Contextual factors are related to the reasons why professionals have never sought this area of work, while the factor of change would imply an adaptation to a new reality. “*I don’t think anybody goes out and intentionally decides they’re going to become a coach for swimmers with a disability*” [27].

#### 3.5.2. Perception of Training Athletes in Adapted Sports

Coaches who have prior knowledge of AS prefer to train in this specific area (positive bias). The selected studies refer to the neutral bias, which refers to trainers who have never thought about the AS topic, yet take on any challenge without reservation, “*Well, I kind of didn’t really think about it all that much before I did it, so when you talk about reservations, I wasn’t really all that conscious of them*” [34].

However, coaches do not only discriminate against positive factors in training athletes with AS, there are also several negative aspects associated with it: negative prejudice, stigma and lack of knowledge. Negative prejudice is seen as difficulties and barriers in exercising the role of the coach. Stigma is a more common phenomenon in elite coaches, since they are afraid of damaging their coaching career and also due to the fact that AS is still seen as an act of charity *“[...] the way that it was presented was that only nice people would do that. So, you know, hard-nosed, ambitious coaches weren’t going to do coaching with disabilities, nice people do that*” [34]. The lack of knowledge is motivated by the lack of social interaction between people with and without disabilities, which causes a lack of knowledge about training, the morphological functioning of athletes, and difficulties in using appropriate terminology.

#### 3.5.3. Development of the Adapted Sports Coach’s Learning

This topic includes the procedural phases that the ASC goes through in the construction of learning and, consequently, the development of their career. In this context, the ASCs develop their knowledge through three paths: theoretical, practical and reflection processes. At a theoretical level, the learning is done through formal procedures. At a practical level this is associated with non-formal and informal contexts. Regarding the non-formal perspective, this is considered as a facilitator of practical learning as it is acquired through participation in congresses, training camps and internships. As for the informal perspective, it is an essential help in the training, as it comes from the learning acquired with the tutors, and in the case of AS we have to consider the knowledge acquisition of athletes, parents, a multidisciplinary team, observation of the training and game as well as the observations of other coaches. The last way corresponds to the reflection, which aggregates three aspects common to all coaches and one specific one in the case of AS. In the general perspective there is the educational reflection, which allows the integration of formal learning contexts and the adaptation to the specificity of this sports area. The reflection with the peers brings together reflective processes with all the agents of the training, namely with the other trainers and athletes, whose interaction may occur before, during and after the training, and is manifested through observation processes (verbal or non-verbal). Reflection in depth summarizes the integration of new knowledge/learning that leads to the transformation of working contexts, i.e., “*coach talked about watching, questioning and then synthesizing important data before making changes to an athlete’s training program or equipment*” [33]. For the ASC, consideration should also be given to adapted thinking, which consists of integrating all learning content and adapting it to meet the specific needs of both athletes and sport.

The research included considers that there are five stages in the construction of the ASC’s career, which complete their journey from the initial stage to the carrying out of the task. At the beginning of the career, it is fundamental to develop the concept of learning to teach, which includes the period of the first contact with the AS, in order to start learning to train in a 2nd stage and which corresponds to the beginning of the specification in the area of training, as well as in the choice of the modality in which you want to develop the work. In the 3rd phase concerning the learning with the mentors, field work begins, guided by trainers with more professional experience, called mentors. The 4th phase is for trainers who have already acquired enough theoretical and practical knowledge to become mentors. The last phase corresponds to the globalization of all stages of the career, allowing lifelong learning.

Career progression is made through the development of the coach in variables such as coach characteristics, training and competition, these being similar for all coaches. For the ASC, contextual factors are also essential for this whole process of evolution. The trainer’s characteristics are related to basic training/experience, trainer’s attributes, and training style. As far as training is concerned, it is important to consider its structure and the way in which the trainers implement the routines, how they set goals and make changes to the plan initially foreseen. During competition it is essential to analyze the key tasks that the trainer must perform before, during and after competition, always considering the particular characteristics of each athlete. Within the AS, there is still to consider the contextual aspects, which means that the reflection must include adaptive references in order to meet the needs of each athlete: “*I take a lot of time thinking about how this piece of equipment, training plan or the temperature outside is going to affect my athletes, sometimes the same, but often differently from the able-bodied program. Not better, or worse, just different*” [33].

#### 3.5.4. Adapted Sports Coach Reality

The reality of the trainer is related to the context that the ASC faces throughout their professional career and this is marked by two poles, the positive and the negative. The positive reality is associated with two benefits: the intrinsic benefit, which corresponds to personal and professional development, and the extrinsic benefit, which highlights interpersonal relationships (parents and athletes). The negative reality comes from 4 factors, which can condition the participation of athletes with disabilities: those of a practical origin (operational and physical), those based on knowledge, psychosocial factors and those related to funding “*The lack of funding provided to athletes with a disability was a common theme reported by the coaches, and one which they felt negatively impacted their capacity to coach to the best of their ability*” [34].

### 3.6. Analytical Subjects

#### 3.6.1. Difficulties of the Adapted Sports Coach

The ASC faces a series of obstacles and difficulties throughout their career: social difficulties; learning difficulties and difficulty in performing the function. As for social difficulties, these are related to stigma in choosing the area of AS, social stigma and financial issues. Stigma in choice is caused by a lack of information and knowledge about sport for disabled people, leading to misconceptions about performing the function. Social stigma arises from society’s general lack of knowledge about disability, leveraged by a lack of educational investment to explain the phenomenon. Finally, funding issues are one of the main obstacles to growth and evolution, as it is deeply unbalanced when comparing AS with Regular Sport.

The learning difficulties of the ASC are a transversal obstacle at all stages of the career, since there is evident difficulty in gathering information about AS. Research on the topic of AS is scarce, which influences the creation and the production of new knowledge. On an academic level, we can find gaps in school subjects about the AS. Therefore, the selected contents only focus on generic issues and on the awareness of that reality. When it comes to the coach’s training and if we take into account the program and the workload that is dedicated to the AS subject, we can see that it is insufficient and that the specific training which tackles the adapted modalities are almost non-existent. The lack of resources previously described forces the ASC to learn through trial and error and to look for alternative sources of knowledge. Several obstacles can manifest when performing their task. In the area of AS, coaches carry out their role not only in the sport aspect, but they also have to perform their task along with handling additional issues, which are created by the lack structure of technical support. During practice there is a lack of adapted equipment, not enough available time to practice and, on many occasions, it is difficult to include athletes with and/or without disability in the same practice. When it comes to the logical aspects there are limitations regarding transportation to the competitions, the lack of access in sporting venues and in the locations for accommodation. Therefore, these aspects create fatigue and recovery time problems in athletes.

#### 3.6.2. Learning Opportunity in Adapted Sport

AS focuses on the myriad possibilities and opportunities to obtain the essential knowledge to produce learning, since it is important for the ASC to know the global reality of the athlete. Knowing this, the articulation between the ASC and the multidisciplinary team who follows the athlete is imperative, in order to allow the coach to obtain significant information about the necessary adaptations for the benefit of the athlete. From the point of view of practical learning, this process allows the ASC to gather new knowledge, create new methodologies, and increase their ability for adaptation and creativity to meet the athletes’ needs.

In AS, athletes have a crucial role in learning, since they are the ones who integrally are aware of their own functional limits. For these athletes, coaches represent great references, as sport is an ally in the changes in their lives, allowing for greater autonomy and social inclusion. The opportunities can also come from a greater interconnection between AS and Regular Sports in a way that they take full advantage of the resources of both.

#### 3.6.3. Particularity in Adapted Sports

For the AS, specificity, particularity and adaptation are key concepts in the building of a fair and competitive environment for all participants. The ASC requires a thorough understanding of the complexity of this athletic subsystem in order to maximize the end result of their work. Therefore, each modality presents its own features in how it manages the disability typology and can also show specificities according to the athletic type.

## 4. Discussion

This study aimed to synthesize and understand the qualitative empirical evidence previously published regarding the career development of ASCs. Besides, the present review also had as a goal the development of a conceptual model for the training and career path of the ASC. The conceptual model was established through the production of a thematic synthesis. The descriptive topics were extracted from the results of the selected studies and establishing the analytical topics, to achieve the study goals. In formal terms, the model was conceived using four descriptive topics and 12 descriptive subtopics, three analytical topics and eight analytical subtopics in order to advance the literature related to the object of study of the present review.

The establishment of this synthesis is an important element for the conception of an efficient training model in the context of adapted sports, which would allow coaches to adapt training processes to the specific needs of their athletes [10,11].

Based on the quality of the evidence, eight studies were considered of high quality and only one of medium quality. All the studies were of a qualitative character. This investigative design can lead to conclusions based on tendentious and exaggerated interpretations [35].

The systematic review uses a strict methodology in the process, including a thorough research of the literature, applying inclusion criteria and selecting the explicitly outlined studies. The collection of the selected studies in the investigation was elaborated through the extraction of data and their subsequent analysis, and the whole process was carried out independently by three investigators. The disagreements were worked out by consensus. Even though the strategy for the literary search was executed using a comprehensive process, we could still detect a publishing bias regarding the selected studies. In which case the research strategies could be insufficient and could also compromise the results [36]. Since all of the studies were written in English, not taking other languages into consideration, and since the sample mainly consisted of elite coaches, there could have been biased predisposition.

The discoveries made in the systematic review are aligned with many current worries in the AS area, where limitations in the scientific research are noticeable, namely about practice and the coach’s role [9,37]. For the coaches who intervene in the AS area, knowledge becomes essential since they have to understand both the sport and the typology of the athlete [36], which means that, in order for the coach to adapt the process and program of the practice to the level of functionality of each individual athlete, these factors are pivotal and cannot be underrated [2]. We can conclude that the ASC faces additional challenges and obstacles. These difficulties are reflected in particular in the gathering of specific information, either on the modalities or on the understanding of the athletes’ disabilities. This procedure leads to trial-and-error coaching, therefore affecting the athletes [28]. The need to improve the ASC’s training is evident, especially in the development of learning situations [27,31]. The combination of the formal, non-formal and informal learning sources is fundamental for learning quality optimization [37]. It is important to note that the tutorials, which promote reflection and an exchange of experiences in pairs, are essential elements that provide the ASC with practical strategies for problem solving [6].

Bentzen et al. [10] indicated that the development of formal education for adaptive sports coaches should be based on empirical research. Furthermore, they state that scientific evidence would help the overall progress of the sport context for disabled athletes. Therefore, further research is needed to better understand the definition of coaching effectiveness in this context. This research aims to further deepen the knowledge of adapted sports coaches in order to generate a specific body of knowledge that is different from that of other coaches.

The rearrangement of the training programs in the AS area are essential so they can become effective and efficient according to the coaches’ needs [38]. Regarding their training, it was suggested it should include practical activity programs, specificities of related contents with the adapted modalities, and observation in a real context [2]. In formal terms and from the academic training viewpoint, the encouragement of research that connects theory and practice is viewed as necessary for the scientific development of the area [39].

Recommendations for future research can be characterized by sectors: methodological deficits, sample characteristics and AS specificities. In relation to the methodological deficits, the inclusion and combination of various research methods were suggested. In regards to the sample characteristics, it was proven to be important to include other sport agents, political decision-makers, disabled coaches, coaches with little experience and women coaches. Lastly, regarding AS specificities, it is important to deepen knowledge of both practice and coach. This way the identification of the ASC’s typology becomes essential, encompassing problematics such as planning, decisions and training, as well as the leadership method used during the process of practice.

## 5. Conclusions

The systematization was made operative with the results of the included studies, where it was perceptible that the path of the ASC is divided into distinct phases that integrate each other and are related to the learning progression in the professional training of the coach. The path of the ASC possesses some particularities when performing the task, making the articulation of the deepened knowledge of the disability typology and the specific adaptations of the modalities, a key element. These suppositions become essential in matching practice to the athlete, optimizing their performance in a specific and particular context.

One of the challenges for AS is the lack of knowledge and financial investment which creates difficulty for the professionals. On the other hand, this area also involves great learning potential, since coaches have to become more creative and innovative in the performance of their task.

To summarize, it is essential to underpin the professional and learning path in the AS area with the development of scientific investigations, in order to solidify the coaches’ knowledge and to allow the athletes to progress.

## Figures and Tables

**Figure 1 ijerph-18-06608-f001:**
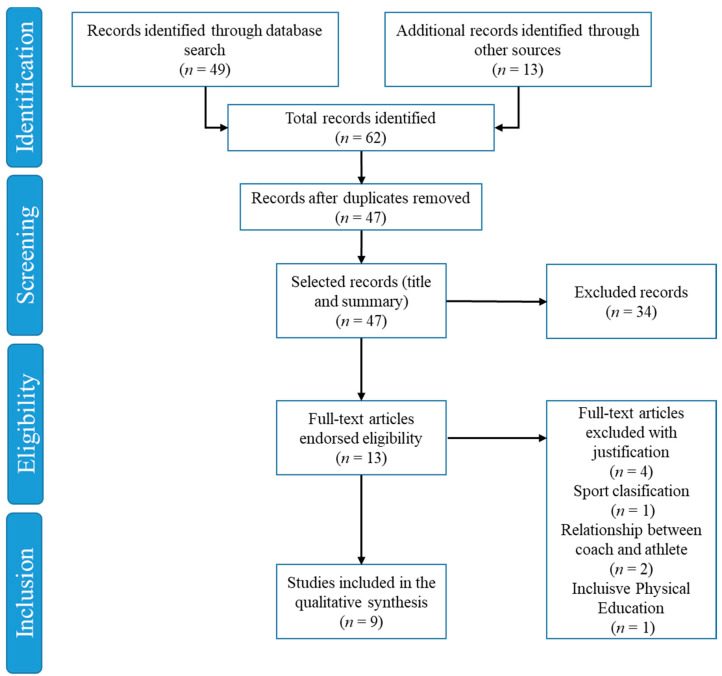
Study selection process using the PRISMA flowchart.

**Figure 2 ijerph-18-06608-f002:**
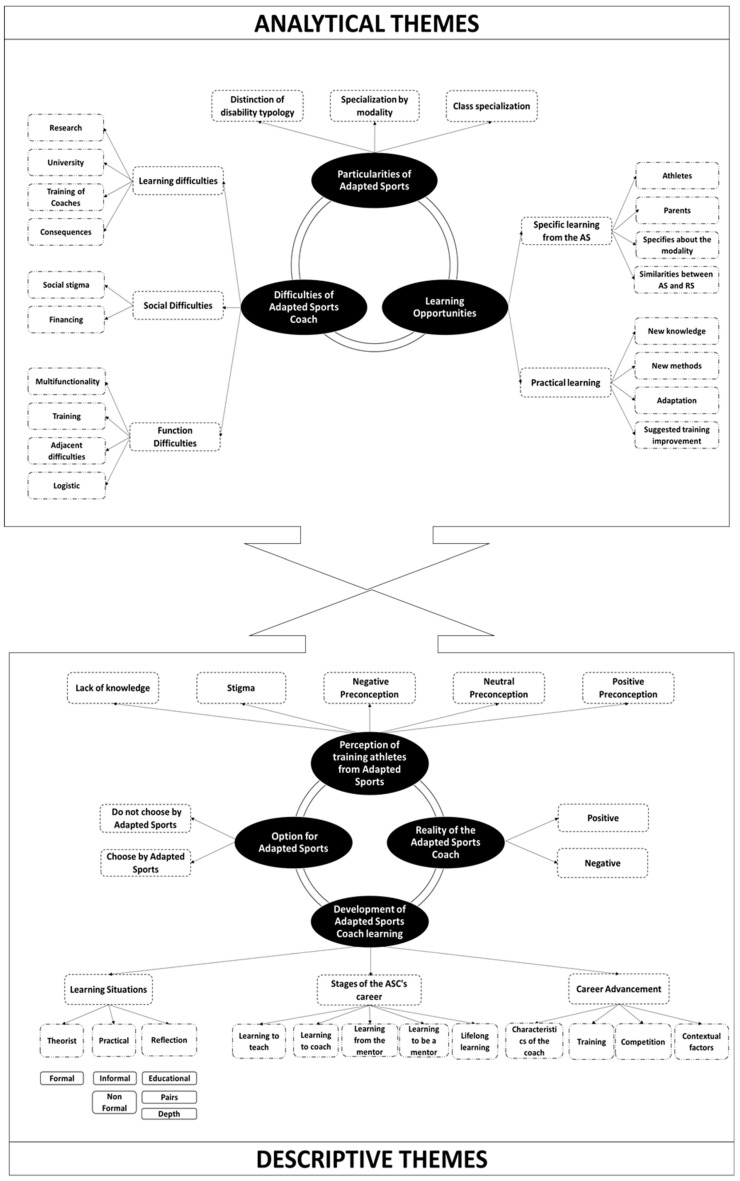
Conceptual Model of Career Development of Adapted Sports Coaches.

**Table 1 ijerph-18-06608-t001:** Development of a Search Strategy and Selection of Eligibility Criteria Based on the SPIDER Acronym.

SPIDER	Description
Sample	Adapted Sports Coach
Phenomenon of Interest	Identify the process of the coaches’ training, as well as their knowledge and skills in the exercise of their role
Design	Interpretative Studies
Evaluation	ASC Career Development, Importance of Reflection on the ASC; Perception of the Reality of AS is related to the theme of knowledge.
Research type	Qualitative

**Table 2 ijerph-18-06608-t002:** Description of the characteristics of the included studies.

Author/Reference/ Country	Design	Goal	Characteristics of the Studies	Data Collection	Data Analysis
Cregan et al. [27], Canada	Qualitative	Examining career development and knowledge of adapted swimming coaches	*n =* 6 (M) TC: Experienced SC: Individual TD: Visual and Motor	Structured Interview	Units of meaning; Identifiers (tags); Properties and Categories
Douglas et al. [28], Canada	Qualitative (Case study)	Understanding the development of the career and learning paths of the Paralympic coaches	*n =* 6 (M/F) TC: Experienced SC: Individual and Team TD: Not Specified	Semi-structured Interview	Thematic Analysis
Duarte et al. [29], Canada	Qualitative (Life stories)	Athlete’s/ASC’s life history	*n =* 1 (F) TC: Experienced SC: Individual TD: Visual and Motor	Rappaport Test; In-depth interview; analysis of personal documents	Thematic Analysis
Fairhurst et al. [30], Canada	Qualitative	Explore the perceptions of the Paralympic coaches about their educational and learning experiences, formal and informal.	*n* = 6 (M) TC: Experienced SC: Individual and Team TD: Not Specified	Semi-structured Interview	Thematic Analysis
MacDonald et al. [2], Canada	Qualitative	Understanding how coaches of mentally disabled athletes acquire their knowledge	*n =* 45 (F/M) TC: Experienced and Beginners SC: Not Specified TD: Intellectual	Structured Interview	Thematic Analysis
McMaster et al. [31], Canada	Qualitative	Examine the learning sources and experiences of the ASCs	*n* = 5 (M/F) TC: Experienced and Beginners SC: Individual and Team TD: Not Specified	Semi-structured Interview	Thematic Analysis
Taylor et al. [32], Canada	Qualitative (Case study)	Explore the learning evolution throughout an ASC’s life course	*n* = 1 (M) TC: Experienced SC: Individual TD: Motor	In-depth interview	Thematic Analysis
Taylor et al. [33], Canada	Qualitative	Understanding through a reflective process the learning moments that helped them carry out their task	*n =* 4 (M) TC: Experienced SC: Individual and Team TD: Not Specified	Semi-structured Interview	Thematic Analysis
Wareham et al. [34], Australia	Qualitative	Holistic view of the experiences and challenges of the ASCs	*n* = 12 (M/F) TC: Experienced SC: Individual and Team TD: Motor and Visual	Semi-structured Interview	Thematic Analysis

**Table 3 ijerph-18-06608-t003:** Quality Appraisal of the Included Studies.

	Aims	Method	Design	Recruitment	Data collection	Relationships	Ethical issues	Analysis	Findings	Value of Research	Rating Category Quality
Cregan et al. [27]	Yes	Can’t tell	Yes	Yes	Can’t tell	Yes	No	Yes	Yes	Yes	medium quality
Douglas et al. [28]	Yes	Yes	No	Yes	Can’t tell	Yes	Yes	Yes	Yes	Yes	high quality
Duarte et al. [29]	Yes	Yes	Yes	Yes	Yes	Yes	Yes	Yes	Yes	Yes	high quality
Fairhurst et al. [30]	Yes	Yes	Yes	Yes	Yes	Yes	Yes	Yes	Yes	Yes	high quality
MacDonald et al. [2]	Yes	Yes	Yes	Yes	Yes	Yes	Yes	Yes	Yes	Yes	high quality
McMaster et al. [31]	Yes	Yes	Yes	Yes	Yes	Yes	Yes	Yes	Yes	Yes	high quality
Taylor et al. [32]	Yes	Yes	Yes	Yes	Yes	Yes	Yes	Yes	Yes	Yes	high quality
Taylor et al. [33]	Yes	Yes	Yes	Yes	Yes	Yes	Yes	Yes	Yes	Yes	high quality
Wareham et al. [34]	Yes	Can’t tell	Yes	Yes	Yes	Yes	Yes	Yes	Yes	Yes	high quality

## Data Availability

Not applicable.

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
