# Peer review of "Career Development of Adapted Sports Coaches: Systematic Review of Qualitative Evidence Literature"

_ijerph, 2021, doi:10.3390/ijerph18126608_

Round 1
Reviewer 1 Report
The topic is interesting and provide some useful information regarding the Adapted Sports career development and the Authors propose a conceptual model on the training and career path of Adapted Sports Coach. The appropriateness of study design, the methodology and the clearness of presentation of the work are good.
Some very minor revisions are required.
Abstract, line 18. Please define the acronym ASC
Introduction lines 34, 42, 48, 53, 57. Cite Authors according to the norms.
Lines 56- 59 The sentence is not clear; in addition, the Authors should specify what 3 + 1 Cs mean.
The first part of the flow chart is not clear. A box is missing before “Records after duplicates removed of duplicates“. This sentence has something wrong.
Table 2 Please define the acronyms TC, SC and TD.
Author Response
Dear Reviewer,
We have carefully considered all considerations in the document provided by you. Enclosed you will find our detailed answers to your inquiries.

Reviewer 2 Report
Abstract
The acronym ASC on line 18 should be avoided or, in any case, its meaning should be indicated in parentheses after Adapted Sports Coaches (ASC) on line 12.
Introduction
It would be convenient to expand the introduction a little further by describing in greater depth similar reviews to the present work, which have been carried out previously. More specifically, the studies by Hagele, Lee and Porrette (2015), that of Bloom, Falcao and Caron (2014), the 3 + 1 Cs model, and especially that of Bentzen, Alexander, Bloom and Kentta (2020), which it is defined as a “scoping review” carried out between 1991 and 2018 and it is not clear whether it is a systematic review with methodological and substantive characteristics similar to the one carried out in the present study.
At a formal level, somewhere should be specified what the acronym ASC stands for.
Eligibility method/criteria
In Table 1 or elsewhere in the text, we note that the dates of the search, the start date and the end date of the search are not clearly identified. Bentzen, Alexander, Bloom and Kentta (2020) carried out a “scoping review” between 1991 and 2018. Is it the same review carried out as of 2018? This aspect should be well clarified. Above all, justify the variants or differential characteristics of this review because, if it is identical to the previous one, the period of time in which the search is carried out is very short.
Discussion / Conclusions
If the present study is a continuation of that of Bentzen et al (2020), comparisons should be made between the results of both studies, and the novelties that have been found should be emphasized.
Author Response

(The authors gave the same response as above.)

Reviewer 3 Report
Thank you for the invitation. The study involves an interesting topic; however, I would like to make some comments.
The introduction does not provide enough information about adapted sport and their coaches. The literature presented is mainly related to sports in general and there is no specific information about the topic of the article.
The eligibility criteria are not clear. It is not possible do reproduce the study because the searching procedures are not clearly described.
Author Response

(The authors gave the same response as above.)

Reviewer 4 Report
In this systematic review, the authors developed conceptual model on the training and career path of Adapted Sports Coach. They included four descriptive subjects and three analytical subjects. Overall, this review is interesting, and may be of practical use.
One minor concern:
-About the difficulties for the ASC, is there any efficient strategy can be applied?
Author Response

(The authors gave the same response as above.)

Reviewer 5 Report
Dear Authors:
The reason for conducting a systematic review is not just to search for the most relevant references. According to my view, the work is processing, it means is still not finished. It must be completed or elaborated again the next parts of the paper: i) the introduction, ii) table III, iii) the discussion and iv) the conclusions.
Table III is completely irrelevant. Authors should aim to include more than 'yes' or 'Can't tell' in Table III. This indication is essential, because it would have a great impact on the rest of the article.
Author Response

(The authors gave the same response as above.)

Round 2
Reviewer 3 Report
Thank you for addressing my comments
Author Response
Thank you very much for your contributions
Reviewer 5 Report
Dear Authors: 1. What kind of valuabe information could readers get from table 3? 2. The english is not enough appropriate. 3. The Authors consider the expression 'Can't tell' in the same table 3 is appropriate? 4. To be honest, i dont think 'Discussion' is enough elaborated. Do you?Author Response
Dear Reviewer,
We have carefully considered all considerations in the document provided by you. Enclosed you will find our detailed answers to your inquiries.
Thanks for reviewing our manuscript.
